# Radical Hysterectomy After Neoadjuvant Chemotherapy for Locally Bulky-Size Cervical Cancer: A Retrospective Comparative Analysis between the Robotic and Abdominal Approaches

**DOI:** 10.3390/ijerph16203833

**Published:** 2019-10-11

**Authors:** Chia-Hao Liu, Yu-Chieh Lee, Jeff Chien-Fu Lin, I-San Chan, Na-Rong Lee, Wen-Hsun Chang, Wei-Min Liu, Peng-Hui Wang

**Affiliations:** 1Department of Obstetrics and Gynecology, Taipei Veterans General Hospital, Taipei 112, Taiwan; chliu12@vghtpe.gov.tw (C.-H.L.); cschen9@vghtpe.gov.tw (I.-S.C.); nllee@vghtpe.gov.tw (N.-R.L.); whchang@vghtpe.gov.tw (W.-H.C.); 2Department of Obstetrics and Gynecology, National Yang-Ming University, Taipei 112, Taiwan; 3Department of Obstetrics and Gynecology, Taipei Medical University Hospital and Taipei Medical University, Taipei 110, Taiwan; d119097012@tmu.edu.tw; 4Department of Statistics, National Taipei University, Taipei 104, Taiwan; cflin@gm.ntpu.edu.tw; 5Department of Orthopedic Surgery, Taipei Municipal Wanfang Hospital, Taipei Medical University, Taipei 116, Taiwan; 6Department of Nursing, Taipei Veterans General Hospital, Taipei 112, Taiwan; 7Institute of Clinical Medicine, National Yang-Ming University, Taipei 112, Taiwan; 8Department of Medical Research, China Medical University Hospital, Taichung 440, Taiwan; 9The Female Cancer Foundation, Taipei 104, Taiwan

**Keywords:** abdominal radical hysterectomy, bulky, cervical cancer, neoadjuvant chemotherapy, outcome, robotic radical hysterectomy

## Abstract

Radical hysterectomy (RH) is the standard treatment for early stage cervical cancer, but the surgical approach for locally bulky-size cervical cancer (LBS-CC) is still unclear. We retrospectively compared the outcomes of women with LBS-CC treated with neoadjuvant chemotherapy (NACT) and subsequent RH between the robotic (R-RH) and abdominal approaches (A-RH). Between 2012 and 2014, 39 women with LBS-CC FIGO (International Federation of Gynecology and Obstetrics) stage IB2–IIB were treated with NACT-R-RH (*n* = 18) or NACT-A-RH (*n* = 21). Surgical parameters and prognosis were compared. Patient characteristics were not significantly different between the groups, but the NACT-R-RH group had significantly more patients with FIGO stage IIB disease, received multi-agent-based NACT, and had a lower percentage of deep stromal invasion than the NACT-A-RH group. After NACT-R-RH, surgical parameters were better, but survival outcomes, such as disease-free survival (DFS) and overall survival (OS), were significantly worse. On multivariate analysis, FIGO stage IIB contributed to worse DFS (*p* = 0.003) and worse OS (*p* = 0.012) in the NACT-A-RH group. Women with LBS-CC treated with NACT-R-RH have better perioperative outcomes but poorer survival outcomes compared with those treated with NACT-A-RH. Thus, patients with FIGO stage IIB LBS-CC disease might not be suitable for surgery after multi-agent-based NACT.

## 1. Introduction

Surgical treatment for cervical cancer (CC) has been discussed intensively owing to recent advances in minimally invasive surgery (MIS). The technical feasibility of minimally invasive radical hysterectomy (MIS-RH) has been previously described in numerous reports [1,2,3,4,5], but there are increasing concerns regarding the oncological outcomes, especially on the basis of the results of the recent prospective randomized trial, i.e., the international laparoscopic approach to cervical cancer (LACC) trial [6,7,8,9,10]. Conventional abdominal RH (A-RH) showed better disease-free survival (DFS) and overall survival (OS) than MIS-RH [8,9], but the current evidence is still debatable [11,12,13,14,15]. The reason is probably because MIS-RH includes both conventionally laparoscopic RH (L-RH) and robotic RH (R-RH), and the techniques needed for conventional laparoscopic surgery are much more advanced and complicated than those for robotic surgery. L-RH requires more time for surgeons to develop the essential level of skill to perform the procedure effectively and has a longer learning curve. Many studies have compared the feasibility of these two different MIS techniques for treating different types of gynecological diseases, regardless of the malignancy status. Unsurprisingly, the results favor the robotic approach [16,17,18,19,20,21,22]. Therefore, when most MIS-RH procedures in the LACC trial were performed via conventional L-RH (84%) and contributed to poorer survival outcomes [9], one could argue that L-RH is not regularly performed in most cancer centers and the current practice and trend favor R-RH in many countries [12], which is also apparent in Taiwan [23,24].

It is also debatable whether RH can be performed instead of concurrent chemoradiotherapy (CCRT) for the management of locally advanced CC. In addition to the well-known criteria such as FIGO (International Federation of Gynecology and Obstetrics) staging, tumor size is among the most concerning factors with an increased possibility of therapeutic failure, and subsequently decreased DFS and OS, regardless of whether surgery was performed [25,26,27,28,29]. Many strategies have been suggested to overcome this limitation, and neoadjuvant chemotherapy (NACT) has been introduced for treating various types of malignancies [29,30,31,32,33,34] and locally advanced CC [28,29,30]. The main purpose of NACT is to reduce tumor size, thereby facilitating subsequent local treatment. Among patients with CC, those who received NACT for FIGO stage IB2/IIA2 CC were highly likely to undergo successful laparoscopic surgery [30]. Other advantages of NACT include the potential of controlling micro-metastatic disease and reducing the need for postoperative adjuvant radiotherapy with/without chemotherapy (CT) [35,36]. To date, only a few studies have evaluated the feasibility and safety of MIS-RH for patients with CC after NACT [37,38,39,40,41,42,43]. In our previous study, we evaluated the therapeutic effect of RH after NACT in patients with locally bulky-size cervical cancer (LBS-CC ≥ 6 cm). Weekly NACT with only cisplatin significantly decreased tumor size (from 6.4 ± 0.5 cm to 4.5 ± 1.4 cm), reduced blood loss (from 930 ± 356 mL to 558 ± 1328 mL, *p* < 0.001), and lowered the immediate complication rate (from 32% to 5.7%) during subsequent A-RH [28]. A meta-analysis of 18 studies with 1785 patients with CC (mainly FIGO stage IB2 to IIB) who were treated with cisplatin-based NACT before RH confirmed that the response to NACT was an indicator of progression-free survival (PFS) and OS [35], suggesting that these patients would have better oncological outcomes if the tumors showed a good response to NACT treatment. Therefore, in this retrospective study, we evaluated and compared the surgical and oncological outcomes of conventional A-RH and R-RH in patients with LBS-CC after NACT.

## 2. Materials and Methods

This retrospective study included patients from two oncological institutions: Taipei Veterans General Hospital (Taipei VGH) and Taipei Medical University Hospital (TMUH) in Taiwan between 2012 and 2014. The study was conducted in accordance with the Declaration of Helsinki, and the protocol was approved by the TMU-Joint Institutional Review Board (project identification code: N201511005). A database search identified 39 patients with histologically confirmed LBS-CC (IB3, IIA2, and IIB, based on the 2018 FIGO staging system for cancer of the cervix uteri [25]) treated with platinum-based NACT followed by radical hysterectomy. Eighteen patients underwent NACT followed by robotic radical hysterectomy (i.e., the NACT-R-RH group) and 21 underwent NACT followed by conventional abdominal radical hysterectomy (i.e., the NACT-A-RH group). The medical charts including age, body mass index (BMI), clinical FIGO stage, pre-treatment tumor size on imaging, intra-operative operative parameters, and length of hospital stay (LOS) were assessed and these data collected. Information about the pathological status was also collected, including pathological tumor size, histology grade, number of resected lymph nodes (LNs), LN involvement, resection margin involvement, parametrial involvement, lymphovascular space invasion (LVSI), and deep stromal invasion (DSI). The pathological specimens were reviewed at each institution by board-certified gynecologic pathologists. The data from both the institutions were pooled into a combined analysis. All the patients who underwent R-RH were informed that the robotic procedure could be converted to A-RH if technological issues occurred.

### 2.1. Neoadjuvant Chemotherapy

All patients with histology-proven CC underwent a complete evaluation before NACT, including a vaginal-pelvic examination, chest radiography, and pelvic computed tomography or magnetic resonance imaging, to evaluate the clinical stage based on the FIGO Clinical Staging of Carcinoma of the Cervix Uteri (2008) and the revised 2018 FIGO staging [25]. Cystoscopy was performed for cases with clinically suspected involvement of the bladder, and colonoscopy was performed for cases with suspected involvement of the rectum. All patients underwent careful evaluation prior to NACT and the following criteria were used: World Health Organization (WHO) performance status of 0–2, adequate bone marrow reserve (absolute granulocyte count ≥ 2000/mL, platelet count ≥ 100,000/mL, and hemoglobin ≥ 10 g/dL); adequate renal, hepatic, pulmonary, and cardiac function; and absence of prior malignant diseases or major surgical illness. Two types of NACT protocols were utilized in this study: platinum-based (40 mg/m^2^) per week for 3 weeks, or taxane/platinum-based (60 mg/m^2^, 40 mg/m^2^) per week for 3 weeks. Surgery was performed at the end of the fourth week. Owing to the retrospective design of this study, the choice of the NACT regimen could not be controlled and it was administered at the discretion of each gynecological oncological surgeon.

### 2.2. Surgical Procedures

All the patients underwent detailed pre-operative evaluations. RH via either approach was type C1 hysterectomy as defined by the updated classification by Querleu and Morrow [44]. Bilateral pelvic lymph node dissection and para-aortic lymph node dissection were performed for all patients in the current study. A single surgeon (W.-M.L.) performed all the robotic surgeries on Da Vinci Si Surgical System^®^ (Intuitive Surgical Inc.^®^, 1020 Kifer Road, Sunnyvale, CA, USA). Central docking between the patient’s legs was used, and all 4 robotic arms were used. One assistant trocar was placed in the left lower abdominal quadrant, and initial access was obtained at the level above the umbilicus by using the closed method.

### 2.3. Pre-Operative Preparation and Post-Operative Care

In both groups, patients received a preventive antibiotic dose of 1 g cefazolin within 30 min before the operation. Elastic compression bandages were applied to the lower extremities to prevent venous thromboembolism. With the patient under general anesthesia, the perineum and vagina were disinfected with povidone-iodine, followed by Foley catheter insertion. The abdomen was disinfected with 0.5% chlorhexidine and tincture of iodine. The operative time was defined from the time of making the skin incision to the completion of skin closure. Estimated blood loss was calculated by the difference in the total amounts of suctioned and irrigation fluids. The postoperative visual analog scale (VAS) pain scores were recorded by nursing staff up to 24 h postoperatively [45,46,47]. On arriving at the recovery room, all patients were immediately administered postoperative opioid (morphine as a 5-mg intravenous injection). Then, postoperative analgesics were administered (morphine as an intravenous infusion with an initial dose of 0.5 mg/h for 1 day and subsequent oral non-steroid anti-inflammatory agent on the next day). Adjuvant therapy was administered according to the presence of risk factors for recurrence considering the final pathology findings such as positive LNs, positive parametrial invasion, positive surgical margins, LVSI, DSI, and tumor size.

### 2.4. Statistical Analysis

Data were represented as the mean and standard deviation (SD) for continuous variables, and number and percentage for categorical variables. The Student *t*-test and Fisher exact test were used to compare the differences between the two groups for continuous variables and categorical variables, respectively. DFS was defined as the duration from the date of surgery to the date of diagnosis of local recurrence or distant metastasis. OS was defined as the duration from the date of surgery to the date of death. The DFS probability and OS probability were estimated using the Kaplan–Meier method. The log-rank test was used to compare differences in the survival probabilities for categorical variables. A univariate Cox proportional hazard model was used to quantify the risk effect on survival for each variable. Possible risk factors associated with survival probability at a significance level of 0.10 or less were included in a multivariable Cox proportional hazard model. A backward variable selection method was used to choose the final model. The level of significance was set at 0.05 for each test before the analysis. All data management and analysis were performed using R 3.5.3 [available at http://www.R-project.org/, R Development Core Team (2018), R: A language and environment for statistical computing. R Foundation for Statistical Computing, Vienna, Austria].

## 3. Results

### 3.1. Clinical Characteristics and Pathological Status

The baseline clinical characteristics were not significantly different between the two groups, including patient age, BMI, comorbidities, a history of previous abdominal surgery, histology type, and tumor size on imaging (Table 1). However, significantly more patients had FIGO stage IIB CC (*n* = 17, 81.0%; and *n* = 4, 22.2%, respectively, all *p* < 0.001) and more patients were treated with multi-agent NACT (taxane/platinum-based) in the NACT-R-RH group than in the NACT-A-RH group (*n* = 19, 90.5%; and *n* = 4, 22.2%, respectively, all *p* < 0.001). The pathological data, including number of resected LNs, LN involvement, resection margin involvement, parametrial involvement, LVSI, and pathological tumor size, were not significantly different between both groups; however, DSI was found in all 18 patients in the NACT-A-RH group (100%) and in 9 patients in the NACT-R-RH group (42.9%; *p* < 0.001).

### 3.2. Perioperative Outcomes and Follow-up Status

Intraoperative findings, such as operative time, estimated blood loss, blood transfusion rate, postoperative VAS score, and LOS were better in the NACT-R-RH group than in the NACT-A-RH group. The incidence of immediate intraoperative complications (bladder, bowel, nerve, and vascular injuries) did not differ between the groups. In the NACT-R-RH group, 2 patients had urinary bladder injury, which was repaired during surgery; in contrast, there were no cases of intraoperative complications in the NACT-A-RH group (Table 2). No conversion to laparotomy was needed in the NACT-R-RH group. The incidence of postoperative complications, such as infection, ileus, and obturator nerve palsy, did not differ between the groups. There were no readmissions due to postoperative complications or re-intervention in either group.

The type of adjuvant therapy administered based on patient risk factors was not significantly different between the groups. The median follow-up time was not significantly different between the groups: 49.6 months (95% confidence interval [CI], 45.3–58.4 months) and 43.7 months (95% CI, 55.8–62.4 months) in the NACT-R-RH and NACT-A-RH group, respectively (*p* = 0.08). At the time of censoring, the sites of recurrences were not significantly different between the two groups. The disease recurrence rate was 47.6% in the NACT-R-RH group and 16.7% in the NACT-A-RH group, without significant difference (*p* = 0.092). Seven patients (33.3%) in the NACT-R-RH group and 1 (5.6%) in the NACT-A-RH group died of disease (*p* = 0.049). The perioperative outcomes and follow-up status are summarized in Table 2.

The worse outcome of patients in the NACT-R-RH group was further reflected by the results on Kaplan-Meir survival analysis. The 5-year DFS rate was 43.3% (95% CI 18.8–65.8) for the NACT-R-RH group and 81.5% (95% CI 52.3–93.7) for the NACT-A-RH group (*p* = 0.0152), as shown in Figure 1. A significantly worse prognosis was noted in patients in the NACT-R-RH group, along with decreased DFS in the 5-year follow-up period. At the end of the first postoperative year, 38% (*n* = 8) of patients in the NACT-R-RH group experienced recurrence, which was significantly higher than the rate of 5.6% (*n* = 1) in the NACT-A-RH group, contributing to the worse prognosis in the 5-year follow-up. The worse DFS also contributed to the worse OS. The 5-year OS rate was 53.9% (95% CI 24.6–76.1) for the NACT-R-RH group and 90.9% (95% CI 50.8–98.7) for the NACT-A-RH group (*p* = 0.0115), as shown in Figure 2. This worse prognosis was significant at the 2-year follow-up; the mortality rate was 52.4% and 16.7% in the NACT-R-RH group and NACT-A-RH group, respectively. To identify the factors that contributed to the worse prognosis, univariate and multivariate analyses were performed. Risk factors associated with worse DFS are shown in Table 3 and Table 4. Univariate analysis for DFS revealed that the surgical approach (R-RH vs. A-RH), FIGO stage (IB3-IIA2 vs. IIB), NACT regimen (taxane/platinum-based vs. platinum-based), LOS, and intra-operative complications might be associated with worse DFS (Table 3). A further multivariate analysis was performed to identify the key factors, and the results showed that FIGO stage IIB and adenosquamous cell carcinoma were independent prognostic factors that shortened the DFS (Table 4). FIGO stage IIB significantly influenced DFS, with a hazard ratio (HR) of 8.71 (95% CI 2.32–48.72, *p* = 0.001). The tumor type of adenosquamous cell carcinoma also contributed to worse DFS (HR 8.54; 95% CI 1.20–63.21; Table 4). On evaluating the risk factors contributing to worse OS, a history of abdominal surgery, FIGO stage IIB, type of the NACT regimen, operative time, LOS, and intraoperative complications were risk factors that decreased OS (Table 5). On multivariate analysis, FIGO stage IIB was identified as the most important prognostic factor contributing to patient death (Table 6). The Kaplan–Meier curve for 5-year DFS comparing FIGO stage IIB and FIGO stage IB3 + IIA2 disease is shown in Figure 3. The Kaplan–Meier curve for 5-year OS comparing FIGO stage IIB and FIGO stage IB3 + IIA2 disease is shown in Figure 4.

## 4. Discussion

The results of the current study demonstrated that NACT-R-RH resulted in significantly better peri-operative outcomes than NACT-A-RH did, with a shorter operative time, lower estimated blood loss, fewer incidences of intraoperative blood transfusion, lower immediate postoperative and 24-h VAS scores, and shorter LOS, without increasing intraoperative complications; these results strongly indicate the feasibility of R-RH in patients with CC. This observation is not new, as a recent meta-analysis suggested that R-RH had surgical benefits for patients with CC than A-RH did [22]. Although the perioperative benefits of R-RH have been well documented, most studies were performed for early stage CC, regardless of whether patients received NACT treatment [12,16,17,18,19,20,21,22,37,38,39,40,41,42,43]. Some studies did not support the benefits of NACT in patients who were scheduled to undergo surgery, especially patients needing MIS [48,49,50]. One of the frequent observations was the desmoplastic effect of the tissue after NACT, making tissue dissection much more difficult during surgery [48,49,50]. In the current study, although 2 patients in the NACT-R-RH group had bladder injury, compared to none in the NACT-A-RH group, we do not believe that this complication is due to the desmoplastic effect of NACT, because all patients in the NACT-R-RH group underwent complete R-RH and no conversion to laparotomy was necessary. Furthermore, intra- and postoperative parameters, such as operative time, estimated blood loss, the need for blood transfusion, and LOS, were significantly better in the NACT-R-RH group than in the NACT-A-RH group; this finding was similar to the findings of previous studies that showed the benefits of NACT, without increasing the difficulty in performing MIS techniques [40,42,51]. In another study, the median estimated blood loss, operative time, and LOS were significantly better after NACT-R-RH than after NACT-A-RH [51]. In a previous study, although the mean operative time was longer in the NACT-R-RH group than in the NACT-A-RH group, other intra-operative parameters, such as estimated blood loss and LOS, were still significantly better in the NACT-R-RH group [50]. In both studies, no significant difference was observed in the recurrence pattern, DFS, and OS during the 3-year follow-up between NACT-R-RH and NACT-A-RH [50,51]. A recent review of many previous case-control and retrospective case series also demonstrated the safety of NACT-R-RH considering the oncological outcomes [52].

The goal of NACT includes downstaging the tumor to improve the radical curability and safety of surgery and inhibit micro-metastasis and distant metastasis [29,30,31,32,33,34,35,36], as well as allow surgeons to perform less radical surgery without compromising oncological safety [52,53]. An early Cochrane review showed a significant decrease in adverse pathological findings after NACT (odds ratio [OR] 0.54, 95% CI 0.40–0.73, *p* < 0.0001 for LN status; OR 0.58, 95% CI 0.41–0.82, *p* = 0.002 for parametrial infiltration), contributing to an improvement in both OS and PFS. These results appear to indicate that NACT may offer benefits over surgery alone for women with LBS-CC, although the effects were less clear for all other pre-specified outcomes [53]. A recent meta-analysis also supported the benefits of decreasing the severity of disease status in patients with LBS-CC treated with NACT, based on the significantly decreased rate of LN metastasis, parametrial infiltration, and recurrence [54]. Thus, NACT is an acceptable and effective procedure for selected patients with LBS-CC, although the analysis did not support advantages in survival (DFS, PFS, and OS) of patients with LBS-CC treated with NACT [54]. In fact, the impact on survival after NACT is still highly debatable, with no consensus about whether it can result in better outcomes [28,29,30,35,36,54,55,56,57,58,59]. Accordingly, the term “acquired treatment response” seems important. A meta-analysis showed varying response rates after NACT for patients with LBS-CC, ranging from 58.5% to 86.5% for the clinical response rate and from 7.5% to 78.8% for the pathological response rate [57]. The clinical response rate predicted favorable DFS (HR 2.36, 95% CI 1.82–3.06) and OS (HR 3.36, 95% CI 2.41–4.69) [57]. Moreover, the pathological response also predicted favorable outcomes such as DFS (HR 3.61, 95% CI 2.0–6.52) and OS (HR 5.45, 95% CI 3.42–8.7) [57]. Another meta-analysis further suggested the importance of an early response, because an early response to NACT is associated with favorable outcomes [58]. However, it is difficult to define “early response”. In the current study, some adverse surgical and pathological parameters seemed to be improved after NACT, although the clinical relevance is unclear. We did not use the same evaluation tools to compare the changes before and after treatment. For example, in the current study, the tumor size decreased by approximately 25% (from 4.6 cm to 3.4 cm in the NACT-R-RH group and from 5.1 cm to 3.6 cm in the NACT-A-RH group, respectively). The size before treatment was calculated using preoperative imaging but the size after treatment was based on postoperative pathological measurement. It is well known that formalin-fixed tissue is significantly smaller than tissue before the process of formalin fixation, with the percentage of shrinkage being approximately 25–30% [28]; therefore, in the current study, tumor size showed almost no response, which might partly explain the worse outcomes.

Another issue in the current study is the use of different NACT regimens for the patients with LBS-CC. In the NACT-R-RH group, more patients received NACT with taxane/platinum-based regimens. It is unclear whether the different NACT regimens contributed to varied outcome in our study. In another study, among women with LBS-CC treated with NACT-RH, taxane/platinum-based regimens had a similar effect on DFS and cause-specific survival compared to non-taxane/platinum-based regimens, irrespectively of the tumor type (squamous cell carcinoma and non-squamous cell carcinoma) [55]. In the current study, the NACT regimen administered was significantly different between the NACT-R-RH and NACT-A-RH groups, and the taxane/platinum-based regimens were administered much more frequently in the NACT-R-RH group. The results from univariate analysis showed that the taxane/platinum-based regimen was associated with worse prognosis, but it may have been influenced by other confounding factors. Further multivariate analysis was performed, and the results did not establish any relationship between different NACT regimens and prognosis. Hence, as the evidence about the beneficial roles of NACT is still not conclusive, further study is warranted.

The worse prognosis of patients with LBS-CC treated with NACT-R-RH demonstrated in the current study led us to re-visit recent publications in the New England Journal of Medicine [8,9]. The LACC trial showed a decreased OS and a 3-fold increase in recurrence in the MIS-RH group than in the A-RH group, both of which were unexpected results [9], suggesting that MIS-RH should be used cautiously. In the current study, FIGO stage IIB was the most important indicator for worse outcomes of patients with LBS-CC treated with NACT-RH. Per the new FIGO staging of cancer of the cervix uteri (2018), invasive carcinoma limited to the cervix with the greatest dimension of ≥2 cm and <4 cm is classified as stage IB2, carcinoma >4 cm as stage IB3, and carcinoma that is limited to the upper two-thirds of the vagina without parametrial invasion and >4 cm as stage IIA2 [25]. This new FIGO staging system of CC emphasizes the importance of tumor size. In a previous study, Wright et al. used the National Cancer Database (*n* = 62,212, data used were of patients treated between 2004 and 2015) to examine the prognostic performance of the 2018 CC staging system; the 5-year survival rate was 91.6% (95% CI 90.4–92.6%) for stage IB1 CC, 83.3% (95% CI 81.8–84.8%) for stage IB2 CC, and 76.1% (95% CI 74.3–77.8%) for IB3 CC [60]. Another group used the data of The Surveillance, Epidemiology, and End Results Program collected between 1988 and 2014 (*n* = 8909) to validate this new FIGO CC staging system and also confirmed that this system was valuable for distinguishing survival outcomes of patients with FIGO stage IB CC [61]. In that study, the HR of cause-specific survival of 2018 FIGO stage IB3 CC and 2018 FIGO stage IB2 CC was 4.07 (95% CI 3.33–4.97) and 1.98 (95% CI 1.62–2.41), respectively, compared to that of 2018 FIGO stage IB1 CC [61]. On multivariable analysis for cause-specific survival in the FIGO stage IB cohort based on 2018 FIGO stage IB2, the HR for 2018 FIGO stage IB3 was 2.06 (95% CI 1.76–2.41) and HR for 2018 FIGO stage IB1 was 0.51 (95% CI 0.42–0.95) [61]. Thus, the 2018 FIGO CC staging system reflected the effect of screening and prevention programs for CC [62,63,64,65,66,67], as the proportion of 2018 FIGO stage IB1 (*n* = 3604, 40.5%) and stage IB2 disease (*n* = 3620, 40.6%) was similar while the percentage of 2018 FIGO stage IB3 disease was only 18.9% (*n* = 1685) [61]. It was anticipated that researchers would find trends in the significantly increasing proportion of 2018 FIGO stage IB1 CC and in the decrease in the 2018 FIGO stage IB3 disease during the study period [61]. The data presented in the current study were consistent with the new 2018 FIGO staging system for CC, and we re-considered the treatment strategy for patients with LBS-CC. Although the cut-off value of 2 cm (2018 FIGO stage IB1 CC) might be applicable and RH could be performed as the MIS procedure [68,69], this cut-off is still debatable [8,9]. Therefore, we strongly agree with the notice of the Society of Gynecologic Oncology (SGO) announced in November 2018, that gynecological oncologists should be aware of the emerging data on MIS for CC so that “thorough discussions can be undertaken with patients and shared decision-making can be used when choosing the surgical approach for RH”. Moreover, the results of the LACC trial, together with institutional data, should be discussed with patients before choosing MIS-RH.

To address the potential risks of MIS for gynecological cancer, many researchers have suggested and hypothesized possible mechanisms that contribute to worse prognosis, thereby assisting in minimizing tumor dissemination during surgery [70,71,72,73,74,75,76,77,78]. Most hypotheses are mostly inconclusive and require larger randomized controlled trials. In addition, MIS-RH is indeed a challenging surgical procedure compared with conventional total hysterectomy [78]. As the surgical technique for MIS-RH varies greatly and is extremely difficult to control for with randomization or multivariate adjustment [69], good training to perform the technique delicately is of paramount importance [79,80,81,82,83].

Several limitations of the current study are apparent due to the unfair comparison between two groups. The first limitation is the small sample size of both groups. The wide use of effective universal screening for cervical cancer (pap smear) in developed countries [62,84,85,86,87] has led to a dramatic decrease in LBS-CC prevalence rate. Moreover, the therapeutic choice for patients with LBS-CC is debated [88]; therefore the patients enrolled in the current study were mainly based on shared decision making, which directly influenced the sample size, resulting in an insufficient power to properly compare PFS and OS between the two types of surgery after NACT. Second, based on the same reason shown above, an additional cost is required for receiving robotic surgery, which is not covered by the National Insurance Health Care System in Taiwan [89,90,91,92,93,94], and the risk of selection bias could not be totally avoided, contributing to a significantly greater number of patients with FIGO stage IIB in the robotic group. This is a well-known limitation of the retrospective design. The effects of both limitations were combined and augmented by the confounding effects. Finally, imaging evaluation was not used at the end of NACT treatment. Therefore, the response of the patients to NACT was uncertain.

Despite these limitations, the current study provided valuable data showing that women with LBS-CC treated with NACT followed by surgery required further evaluation, especially for those patients with FIGO stage IIB. Our study confirmed that the seemingly promising perioperative outcomes in the NACT-R-RH group did not translate into better oncological outcomes. In contrast, they were associated with higher recurrence and mortality rates. Therefore, it is clear that any advanced technology (robotic surgery) in cancer treatment should be testified by survival outcomes, regardless of perioperative outcomes. In addition, the current study provided additional evidence that FIGO stage IIB contributed greatly to a poorer prognosis if NACT-R-RH was performed. Therefore, it is reasonable to suppose that NACT-A-RH might be a more appropriate approach in patients with FIGO stage IB3-IIA2 LBS-CC if the patients plan to undergo NACT along with a subsequent surgical intervention.

## 5. Conclusions

R-RH after NACT in patients with LBS-CC results in better perioperative outcomes but does not contribute to better survival outcomes. In addition, NACT-R-RH is associated with higher rates of recurrence and mortality, and the poor oncological outcomes occur relatively earlier compared to that observed after NACT-A-RH. Patients with FIGO stage IIB LBS-CC should be well informed about the possible inferior survival outcomes before undergoing NACT-R-RH.

## Figures and Tables

**Figure 1 ijerph-16-03833-f001:**
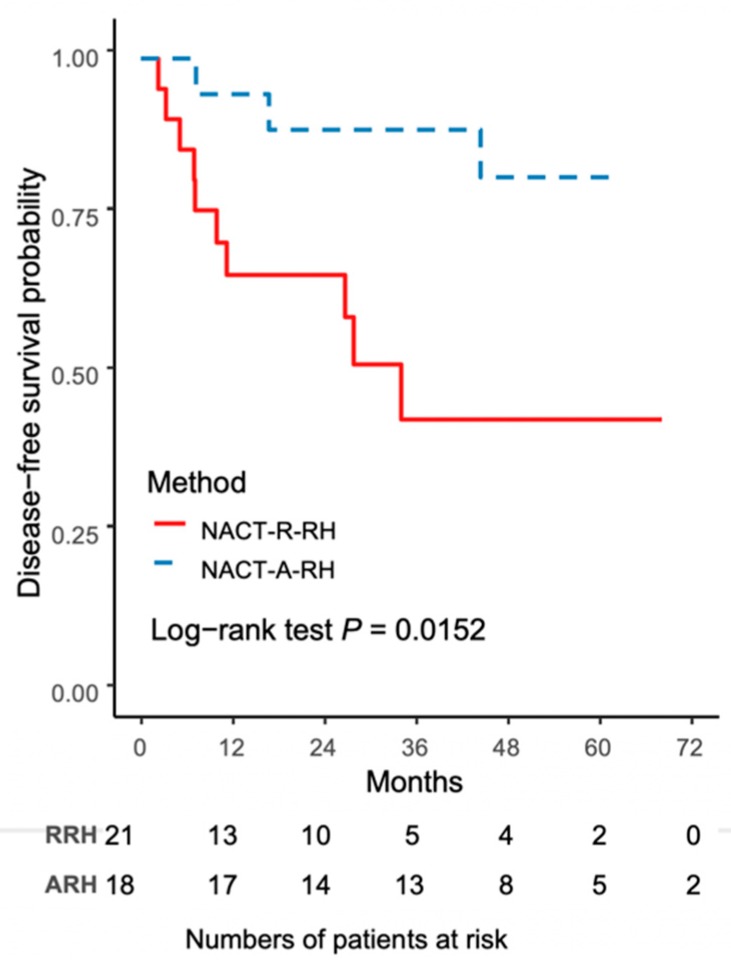
The Kaplan–Meier curve for 5-year disease-free survival (DFS) comparing NACT-R-RH and NACT-A-RH.

**Figure 2 ijerph-16-03833-f002:**
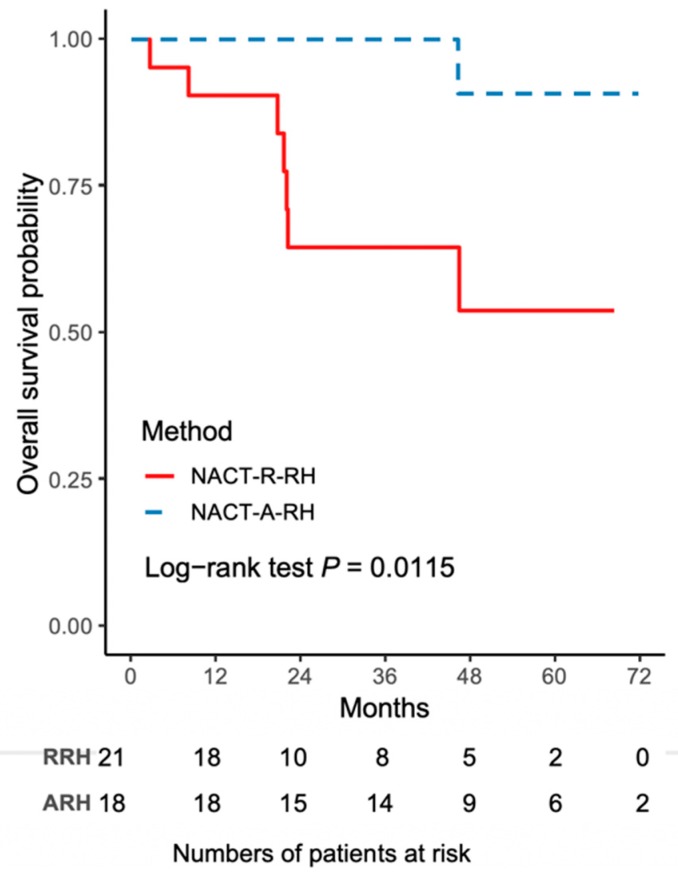
The Kaplan–Meier curve for 5-year overall survival (OS) comparing NACT-R-RH and NACT-A-RH.

**Figure 3 ijerph-16-03833-f003:**
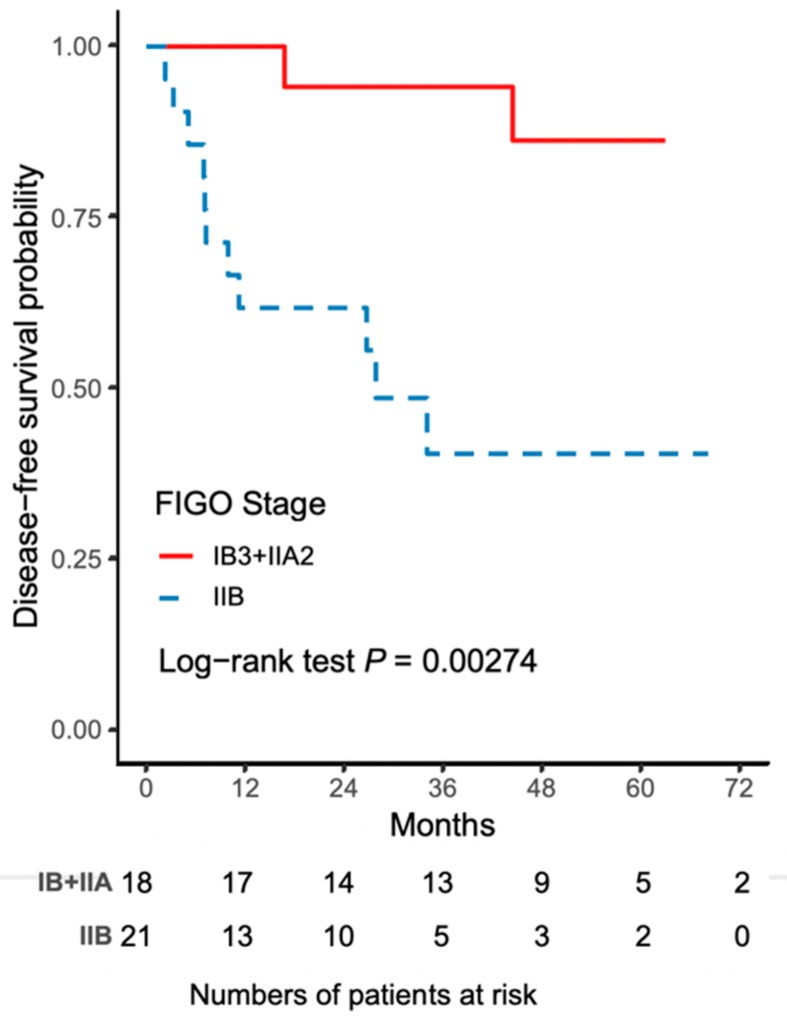
The Kaplan–Meier curve for 5-year disease-free survival (DFS) comparing FIGO stage IIB and FIGO stage IB3 + IIA2 disease.

**Figure 4 ijerph-16-03833-f004:**
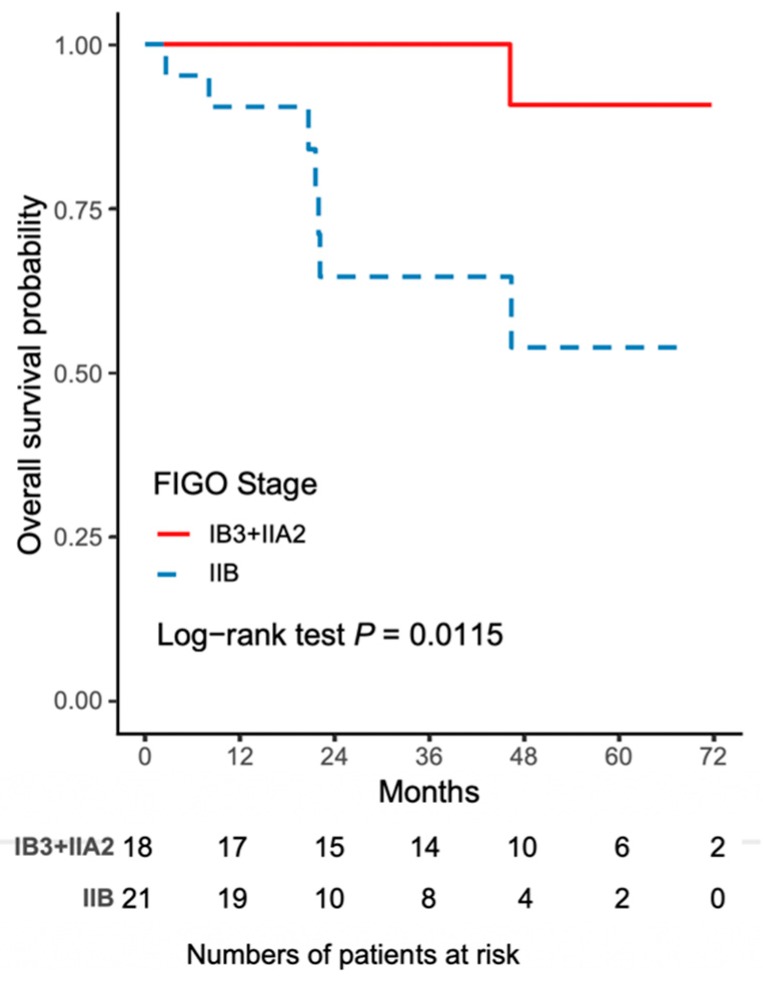
The Kaplan–Meier curve for 5-year overall survival (OS) comparing FIGO stage IIB and FIGO stage IB3 + IIA2 disease.

**Table 1 ijerph-16-03833-t001:** Clinical characteristics and pathological status.

Variable	Category	R-RH	A-RH	*p* Value
Number of patients		21	18	
Age (years)		49.76 (11.61)	50.00 (11.48)	0.949
BMI (kg/m^2^)		23.83 (2.77)	22.48 (2.37)	0.114
Comorbidities	Yes	3 (14.3%)	6 (33.3%)	0.255
	No	18 (85.7%)	12 (66.7%)	
History of abdominal surgery	Yes	9 (42.9%)	4 (22.2%)	0.307
	No	12 (57.1%)	14 (77.8%)	
Tumor size on imaging		4.60 (2.06)	5.05 (1.37)	0.484
Pathological tumor size		3.40 (1.68)	3.61 (1.52)	0.72
Clinical stage (FIGO)	IB3	3 (14.2%)	9 (50.0%)	0.043
	IIA2	1 (4.8%)	5 (27.8%)	0.077
	IIB	17 (81.0%)	4 (22.2%)	0.001
Histology	Adenocarcinoma	8 (38.1%)	2 (11.1%)	0.113
	ASC	0 (0.0%)	2 (11.1%)	
	SCC	13 (61.9%)	14 (77.8%)	
NACT regimen	Platinum-based	2 (9.5%)	14 (77.8%)	<0.001
	Taxane/platinum-based	19 (90.5%)	4 (22.2%)	
Grade (%)	1	1 (4.8%)	0 (0.0%)	0.464
	2	14 (66.7%)	15 (83.3%)	
	3	6 (28.6%)	3 (16.7%)	
Number of LNs resected		27.35 (9.14)	30.28 (8.79)	0.341
LN involvement (%)	Positive	7 (33.3%)	5 (27.8%)	0.979
	Negative	14 (66.7%)	13 (72.2%)	
Resection margin involvement (%)	Positive	7 (36.8%)	3 (16.7%)	0.269
	Negative	12 (63.2%)	15 (83.3)	
Parametrial involvement (%)	Positive	4 (19.0%)	2 (11.1%)	0.667
	Negative	17 (81.0%)	16 (88.9%)	
LVSI (%)	Positive	10 (47.6%)	7 (38.9%)	0.748
	Negative	11 (52.4%)	11 (61.1%)	
DSI (%)	Positive	9 (42.9%)	18 (100.0%)	<0.001
	Negative	12 (57.1%)	0 (0.0%)	

The data are presented as mean (standard deviation) or number (percentage). R-RH: robotic radical hysterectomy; A-RH: abdominal radical hysterectomy; BMI: body mass index; ASC: adenosquamous cell carcinoma; SCC: squamous cell carcinoma; NACT: neoadjuvant chemotherapy; LN: lymph node; LVSI: lymphovascular space invasion; DSI: deep stromal invasion.

**Table 2 ijerph-16-03833-t002:** Perioperative outcomes and follow-up status.

Variable	Category	R-RH	A-RH	*p* Value
Number of patients		21	18	
Operative time (minutes)		195.86 (54.58)	292.50 (59.50)	<0.001
Estimated blood loss (mL)		188.10 (150.75)	941.25 (569.29)	<0.001
Blood transfusion	Yes	0 (0%)	5 (27.8%)	0.015
	No	21 (100%)	13 (72.2%)	
Postoperative pain score (VAS)		2.62 (0.59)	6.33 (0.71)	<0.001
24-h postoperative pain score (VAS)		1.62 (0.50)	3.33 (0.50)	<0.001
Hospital stay (days)		5.95 (3.04)	13.83 (2.09)	<0.001
Intraoperative complications	Yes	2 (9.5%)	0 (0%)	0.49
	No	19 (90.5%)	18 (100%)	
Conversion to laparotomy		0 (0%)		
Postoperative complications	Yes	3 (14.3%)	2 (11.1%)	0.814
	No	18 (85.7%)	16 (88.9%)	
Readmission		0 (0%)	0 (0%)	
Re-intervention		0 (0%)	0 (0%)	
Adjuvant therapy	CCRT	12 (57.1%)	9 (50%)	0.624
	RT	1 (4.8%)	3 (16.7%)	
	CT	4 (19%)	3 (16.7%)	
	None	4 (19%)	3 (16.7%)	
Recurrence	Yes	10 (47.6%)	3 (16.7%)	0.092
	No	11 (52.4%)	15 (83.3%)	
	Distal	4 (19%)	0 (0%)	
	Abdomen	2 (9.5%)	1 (5.6%)	
	Pelvis	4 (19%)	1 (5.6%)	
	Local	0 (0%)	1 (5.6%)	
Disease-specific death	Yes	7 (33.3%)	1 (5.6%)	0.049
	No	14 (66.7%)	17 (94.4%)	

The data were presented as mean (standard deviation) or number (percentage). R-RH: robotic radical hysterectomy; A-RH: abdominal radical hysterectomy; VAS: visual analogue scale; CCRT: concurrent chemoradiotherapy; RT: radiation therapy; CT: chemotherapy.

**Table 3 ijerph-16-03833-t003:** Univariate analysis of disease-free survival.

Variable	Category	Beta	HR	95% CI	*p* Value
Age (years)		−0.033	0.967	(0.913–1.020)	0.225
BMI (kg/m^2^)		−0.074	0.929	(0.746–1.138)	0.484
Comorbidities	No	0	1		
	Yes	−0.624	0.536	(0.104–1.828)	0.343
History of abdominal surgery	No	0	1		
	Yes	1.010	2.747	(0.946–8.202)	0.063
Operative method	R-RH	0	1		
	A-RH	−1.386	0.250	(0.062–0.785)	0.017
Tumor size on imaging		0.016	1.016	(0.736–1.390)	0.920
Pathological tumor size		0.317	1.373	(0.919–1.930)	0.113
Clinical stage (FIGO)	IB3 + IIA2	0	1		
	IIB	1.826	6.211	(1.796–32.389)	0.003
Histology	Adenocarcinoma	0	1		
	SCC	−0.001	0.999	(0.279–5.264)	0.999
	ASC	0.661	1.937	(0.277–13.137)	0.482
NACT regimen	Platinum-based	0	1		
	Taxane/platinum-based	1.155	3.173	(1.005–12.850)	0.049
Grade	1	0	1		
	2	0.013	1.013	(0.129–130.569)	0.993
	3	0.145	1.156	(0.111–155.876)	0.922
Number of LNs resected		0.001	1.001	(0.930–1.074)	0.975
Number of positive LNs		0.227	1.255	(0.903–1.629)	0.157
LN metastasis	Negative	0	1		
	Positive	0.446	1.562	(0.498–4.502)	0.425
Surgical margin	Negative	0	1		
	Positive	0.790	2.203	(0.673–6.953)	0.183
Parametrial invasion	Negative	0	1		
	Positive	0.575	1.777	(0.448–5.485)	0.377
LVSI	Negative	0	1		
	Positive	0.635	1.887	(0.651–5.616)	0.237
DSI	Negative	0	1		
	Positive	−0.967	0.380	(0.129–1.145)	0.084
Adjuvant therapy	None	0	1		
	CCRT	0.697	2.007	(0.422–19.271)	0.411
	RT	−0.607	0.545	(0.004–10.235)	0.698
	CT	1.257	3.514	(0.732–33.913)	0.123
Operative time		−0.007	0.993	(0.984–1.001)	0.079
Blood loss		−0.001	0.999	(0.997–1.000)	0.137
Blood transfusion	No	0	1		
	Yes	−0.422	0.656	(0.071–2.714)	0.606
Hospital stay (days)		−0.121	0.886	(0.777–1.000)	0.049
Intraoperative complications	No	0	1		
	Yes	1.688	5.408	(1.038–18.829)	0.046

HR: hazard ratio; CI: confidence interval; BMI: body mass index; R-RH: robotic radical hysterectomy; A-RH: abdominal radical hysterectomy; SCC: squamous cell carcinoma; ASC: adenosquamous cell carcinoma; NACT: neoadjuvant chemotherapy; LN: lymph node; LVSI: lymphovascular space invasion; DSI: deep stromal invasion; CCRT: concurrent chemoradiotherapy; RT: radiation therapy; CT: chemotherapy.

**Table 4 ijerph-16-03833-t004:** Multivariate analysis of disease-free survival.

Variable	Category	Beta	HR	95% CI	*p* Value
Clinical stage (FIGO)	IB3 + IIA2				
	IIB	2.165	8.710	(2.320–48.723)	0.001
Histology	Adenocarcinoma				
	SCC	−0.243	0.784	(0.212–4.186)	0.743
	ASC	2.145	8.544	(1.200–63.213)	0.034

HR: hazard ratio; CI: confidence interval; SCC: squamous cell carcinoma; ASC: adenosquamous cell carcinoma.

**Table 5 ijerph-16-03833-t005:** Univariate analysis of overall survival.

Variable	Category	Beta	HR	95% CI	*p* Value
Age (years)		−0.022	0.978	(0.924–1.031)	0.420
BMI (kg/m^2^)		−0.044	0.957	(0.757–1.200)	0.707
Comorbidities	No	0	1		
	Yes	−0.468	0.626	(0.121–2.166)	0.487
History of abdominal surgery	No	0	1		
	Yes	1.186	3.275	(1.084–10.553)	0.036
Surgery method	R-RH	0	1		
	A-RH	−1.773	0.170	(0.032–0.594)	0.004
Tumor size on imaging		0.104	1.109	(0.788–1.551)	0.545
Pathological tumor size		0.372	1.451	(0.936–2.139)	0.091
Clinical stage (FIGO)	IB3-IIA2	0	1		
	IIB	2.318	10.158	(2.406–94.007)	0.001
Histology	Adenocarcinoma	0	1		
	SCC	−0.001	0.999	(0.279–5.264)	0.999
	ASC	0.661	1.937	(0.277–13.137)	0.482
NACT regimen	Platinum-based	0	1		
	Taxane/Platinum-based	1.590	4.905	(1.382–25.901)	0.012
Grade	1	0	1		
	2	0.155	1.168	(0.146–151.249)	0.913
	3	0.181	1.199	(0.113–162.179)	0.903
Number of LNs resected		−0.003	0.997	(0.923–1.074)	0.948
Number of positive LNs		0.216	1.241	(0.875–1.638)	0.201
LN metastasis	Negative	0	1		
	Positive	0.184	1.202	(0.374–3.523)	0.744
Surgical margin	Negative	0	1		
	Positive	0.896	2.449	(0.724–8.285)	0.145
Parametrial invasion	Negative	0	1		
	Positive	0.552	1.737	(0.432–5.501)	0.401
LVSI	Negative	0	1		
	Positive	0.609	1.839	(0.612–5.885)	0.275
DSI	Negative	0	1		
	Positive	0.579	1.785	(0.370–17.240)	0.502
Adjuvant therapy	None	0	1		
	CCRT	0.579	1.785	(0.370–17.240)	0.502
	RT	−0.711	0.491	(0.003–9.539)	0.651
	CT	1.304	3.684	(0.737–36.542)	0.119
Operative time		−0.010	0.990	(0.980–0.999)	0.027
Blood loss		−0.001	0.999	(0.996–1.000)	0.070
Blood transfusion	No	0	1		
	Yes	−0.295	0.744	(0.081–3.131)	0.725
Hospital stay (days)		−0.174	0.840	(0.717–0.963)	0.012
Intraoperative complications	No	0	1		
	Yes	1.719	5.577	(1.038–20.996)	0.046

HR: hazard ratio; CI: confidence interval; BMI: body mass index; R-RH: robotic radical hysterectomy; A-RH: abdominal radical hysterectomy; SCC: squamous cell carcinoma; ASC: adenosquamous cell carcinoma; NACT: neoadjuvant chemotherapy; LN: lymph node; LVSI: lymphovascular space invasion; DSI: deep stromal invasion; CCRT: concurrent chemoradiotherapy; RT: radiation therapy; CT: chemotherapy.

**Table 6 ijerph-16-03833-t006:** Multivariate analysis of overall survival.

Variable	Category	Beta	HR	95% CI	*p* Value
Clinical stage (FIGO)	IB3 + IIA2				
	IIB	2.041	7.702	(1.602–75.450)	0.009

HR: hazard ratio; CI: confidence interval; SCC: squamous cell carcinoma; ASC: adenosquamous cell carcinoma.

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
