# Peer review of "Radical Hysterectomy After Neoadjuvant Chemotherapy for Locally Bulky-Size Cervical Cancer: A Retrospective Comparative Analysis between the Robotic and Abdominal Approaches"

_ijerph, 2019, doi:10.3390/ijerph16203833_

Round 1

Reviewer 1 Report

Review for Manuscript ijerph-589283-peer-review-v1

General Comments: Very well designed and written and extremely easy to review. All comments are minor below. However, a major comment that needs further explanation/discussion is the fact that the robotic group had a significantly greater number of patients with higher grade tumors (IIB), which may sway the results towards a worst prognosis using some comparison criteria when in fact the outcome may have been the same for these patients due to a worse tumor grade if these were treated with an open surgical approach.

More Specific Comments:

Title – None

Abstract – None

Introduction – First paragraph, change “those for robotic surgery are” to “those for robotic surgery”

Materials and Methods – First paragraph, change “(LOS) were assessed and collected” to “(LOS) were assessed and these data collected”

Results – None

Discussion – See general comment above.

Conclusions – See general comment above.

Figure and Table Legends – None

Figures and Tables – In the tables, comparisons within rows should have each P value stated. For example, in Table 1, the P value of 0.001 for FIGO IIB is on the same line as FIGO IB3. Therefore, it looks as if the IB3 is significantly different but not the IIB. This should be done for all tables – each comparison has a P value so the presentation is not accidentally misleading.

Reviewer 2 Report

In the present study, the authors identified 39 patients with histologically confirmed locally bulky-size cervical cancer (LBS-CC). Eighteen patients were treated under neoadjuvant chemotherapy (NACT) followed by robotic radical hysterectomy (i.e., the NACT-R-RH group). 21 out of 39 patients underwent NACT followed by conventional abdominal radical hysterectomy (i.e., the NACT-A-RH group). Surgical parameters and prognosis were analyzed to investigate the perioperative outcomes of the RH between the robotic (R-RH) and abdominal approaches (A-RH). This study confirms the R-RH after NACT in patients with LBS-CC results in better perioperative outcomes but does not contribute to better survival outcomes.

The finding provides significant prognostic value, which is also supported by cited studies. Although the study is lack of novelty, the solid and comprehensive analysis make it a good publication. No further questions need to be addressed.                   

Reviewer 3 Report

The authors present a retrospective study on a important topic. Writing is fine. However, it is not clear for me what they want to show.

They compare 18 patients with robotic surgery with 21 patients undergoing abdominal surgery. The groups were quite different compared to tumor characteristics and adjuvant therapy. This makes it very difficult to compare both groups in terms of survival. Even if you do multivariate analysis this is not really reliable. Any patient in this collective is important. Therefore, I encourage the authors to publish their paper. I would suggest to better focus its messages. I would suggest writing it more as a descriptive study instead of a comparative study because the groups are not really comparable.
